# Whole-Genome Sequencing for Identifying Candidate Genes Related to the Special Phenotypes of the Taihu Dianzi Pigeon

**DOI:** 10.3390/ani14071047

**Published:** 2024-03-29

**Authors:** Rui Zhang, Chunyu Mu, Lingling Chang, Xinyue Shen, Zhu Bu, Mingjun Yang, Shengyong Fu, Qingping Tang, Peiyao Liu, Xiaoming Yang

**Affiliations:** 1Institute of Poultry Science, Chinese Academy of Agricultural Sciences Poultry Institute, Yangzhou 225100, China; zrjindy@163.com (R.Z.); muchunyu521@126.com (C.M.); jqscll@163.com (L.C.); shenxy0618@163.com (X.S.); jsbuzhu@163.com (Z.B.); xnfsy@163.com (S.F.); 2Henan Tiancheng Pigeon Industry Co., Ltd., Pingdingshan 462513, China; tianchenggeye@163.com (M.Y.); lpy1003985692@126.com (P.L.); yy2007319@126.com (X.Y.)

**Keywords:** whole-genome sequencing, piebalding, crest, polydactyly

## Abstract

**Simple Summary:**

Pigeons are highly diverse and exhibit a variety of external features, such as a special piebald pattern, crest, and polydactyly. However, the mechanism underlying the formation of these special phenotypes has not been elucidated. We aimed to study the candidate genes associated with the special piebald, crest, and polydactyly traits in the Taihu Dianzi pigeon. Some candidate genes were identified by a selective sweep and GWAS. This result may be important for understanding each trait, which could establish a foundation for the development and utilization of local pigeon resources in China.

**Abstract:**

The Taihu Dianzi pigeon is a breed native to China, and its special piebalding, crest, and polydactyly phenotypes are the result of artificial and natural selection. Here, we analyzed the genetic differences among three kinds of pigeons with different phenotypes at the genomic level. A selective sweep was conducted based on the fixation index (*F_ST_*) and nucleotide diversity (*π*) ratio, and the results revealed that *MC1R* was related to the formation of the distinctive piebalding of the Taihu Dianzi pigeon. Combined with the results of genome-wide association studies, we identified candidate genes associated with the crest (*SMYD* and *STOX2*) and polydactyly (*SLC52A3* and *ANGPT4*). The candidate genes identified in this study and their variants may be useful for understanding the genetic mechanism underlying the special phenotypes of the Taihu Dianzi pigeon. This study provides new insights into the genetic factors that may influence the formation of the special piebalding, crest, and polydactyly characteristics in pigeons.

## 1. Introduction

The pigeon (*Columba livia*) is a common bird worldwide that was deeply loved by Darwin and is extensively described in the book “Origin of Species.” Pigeons are highly diverse and exhibit more trait variation than any other bird [1]; they are known as an ideal model for different investigations, such as those on ecology, behavior, and genetic diversity [2]. Like in other domestic animals, natural and artificial selection have impacted the genetic evolution of pigeons [3]. For several reasons, such as environmental adaptation, communication requirements, people’s preferences, and sports, pigeons have evolved various phenotypic characteristics [4,5]. Due to their gentle temperament and distinct appearance, some pigeons have been collected and cultivated for ornamental purposes for a long time.

In China, pigeon breeding has been a normalized activity since ancient times, during which several kinds of domestic breeds have been selected. Hundreds of breeds exhibit extensive variation in plumage color and pattern [6]. The Taihu Dianzi pigeon is a domestic ornamental breed that originated in the Taihu Lake area of China. It has white plumage covering its body, and its black tail and black spot on the top of the head form a special piebalding. Compared with ordinary pigeons, some Taihu Dianzi pigeons have a crest on their forehead or one more toe. Because of its beautiful appearance and special characteristics (piebalding, a crest and polydactyly), it is a special resource that was generated over hundreds of years and is loved by local people. A pigment pattern that has evolved repeatedly within and among species is called piebalding. Piebalding is characterized by patches of pigmented and non-pigmented feathers, and these plumage patterns are often breed-specific and stable across generations [6]. The crest trait is a specific and widely distributed phenotype in birds, but the shape and physiological characteristics of this trait vary among different species of birds. The crest in chickens consists of feathers on the head that are elongated and upraised and differ between breeds in density, size, and shape [7]. The cerebral hemispheres of partially crested chickens are extruded into the spherical region of the skull, and the anterodorsally part of the skull is expanded into a large spherical protuberance, referred to as a “cerebral hernia” [8], but this is not present in all chickens [9]. The crest cushion of the crested duck consists of soft tissue protuberances covered by feathers and skin [10]. Currently, there is limited research on the crests of pigeons. A previous study on the crest at the back of the pigeon brain showed that the formation of this kind of crest is due to the reverse growth of the head feathers during development, and the authors identified the *EphB2* gene as a candidate for the development of this kind of crest in pigeons [11]. Therefore, it can be inferred that crest formation is a complex process involving different genetic mutations and physiological factors.

In general, fowls have four toes on each foot, while some breeds have more toes, which is known as polydactyly. The four-toed condition of the fowl is the result of the loss of the fifth digit from the typical pentadactyl foot of higher vertebrates. Various researchers have shown that the additional toe in five-toed breeds does not constitute a restoration of the missing fifth digit but rather is due to the development of a new toe on the foot [12]. Studies on polydactyl chickens have shown that polydactyly in chickens of different origins can be caused by different mutations that are closely located in the chicken genome [13]. However, current studies on the mechanism underlying polydactyly development in pigeons are limited. Here, we selected three kinds of pigeon breeds, including two pigeon breeds native to China and an introduced pigeon breed (Figure 1). White Carneau pigeons are bred for commercial use and covered with full-body white feathers. The Tarim pigeon is native to the Tarim Basin in the Xinjiang Uygur Autonomous Region, and the central production areas are the Kashi and Aksu regions. The Tarim pigeons have undergone almost no artificial selection, so their appearance is similar to that of wild pigeons. The above two types of pigeons exhibit neither crests nor polydactyly. This study aimed to uncover the molecular targets that shape the unique morphological features of the Taihu Dianzi pigeon. To do this, we conducted genome-wide selective sweep and genome-wide association studies (GWAS) to identify genomic regions and candidate genes that may explain the distinctive phenotypes.

## 2. Materials and Methods

### 2.1. Ethics Statement

This study was conducted from November 2022 to June 2023 at the Institute of Poultry Science, Chinese Academy of Agricultural Sciences (PI-CAAS), Yangzhou, China. The experimental procedures were approved by the Animal Ethics Committee of the PI-CAAS, and humane animal care and handling procedures were followed throughout the experiment (protocol number: PI-CAAS-2022-10).

### 2.2. Animals and Whole Genome Sequencing

The three breeds included 33 Taihu Dianzi pigeons (DZ1: normal (11); DZ2: crested (11); DZ3: polydactyl (11)), 9 Tarim pigeons (TRM), and 11 white Carneau pigeons (CR) from Jiangsu Weitekai Pigeon Industry Co., Ltd. of China (Jiangyin, China) were included in this study. Total genomic DNA was extracted from blood samples using the CTAB (cetyltrimethylammonium bromide) method. At least 3 μg genomic DNA was used to construct paired-end libraries with an insert size of 300–400 bp using a Paired-End DNA Sample Prep kit (Illumina Inc., San Diego, CA, USA). These libraries were sequenced using the novaseq6000 (Illumina Inc., San Diego, CA, USA) NGS platform at Genedenovo Biotechnology Co., Ltd. (Guangzhou, China).

### 2.3. Variants Identification and Annotation

To identify SNPs and InDels, the Burrows-Wheeler Aligner Reads (BWA _0.7.12) was used to align the clean reads from each sample against the pigeon reference genome Cliv_2.1 (https://www.ncbi.nlm.nih.gov/assembly/GCA_000337935.2 (accessed on 15 January 2023)) with the settings ‘mem 4 -k 32 -M’, -k is the minimum seed length, and -M is an option used to mark shorter split alignment hits as secondary alignments [14]. Variant calling was performed for all samples using the GATK’s Unified Genotyper. SNPs and InDels were filtered using GATK’s Variant Filtration and those exhibiting segregation distortion or sequencing errors were discarded. To determine the physical positions of each SNP, the software tool ANNOVAR (http://www.openbioinformatics.org/annovar/) [15], was used to align and annotate SNPs or InDels.

### 2.4. Population Genetics Analysis

The neighbor-joining (NJ) tree based on filtered SNPs was constructed using MEGA-X software (www.megasoftware.net) (p-distance model) [16]. The population subdivision pattern was preliminarily classified in the principal component analysis (PCA) by the software GCTA (version 1.25.2) [17].

### 2.5. Selective Sweep Analysis

To ensure the accuracy of the analysis, the SNPs with missing rates greater than 20% and minor allele frequency (MAF) less than 1% were first excluded from further analysis. Multiple indexes related to population genetics, including nucleotide diversity (*π*) [18], *π* ratio [19], and fixation index (*F_ST_*) [20], were calculated by the software PopGenome (version 2.2.4) [21] with a sliding window approach. The window size was set to 100 kb and the step size was set to 10 kb. Selective sweep regions were selected according to the interception of 2 indexes, which were *F_ST_* and *π* ratio, with a threshold of the top 5% level. All related graphs were drawn by R scripts.

### 2.6. Genome-Wide Association Analysis

Genome-wide association mapping was implemented in TASSEL [22] v.5.2.54, using mixed linear model accounts for both population structure and kinship as fixed and random effects, respectively (MLM(QK)). The Bonferroni correction threshold (0.01/marker number) was used to identify significant associations. Candidate genes (CAGs) located within the 50-kb region upstream or downstream of significant associated makers were identified.

## 3. Results

### 3.1. Genome Resequencing of Five Pigeon Groups

Genome sequencing yielded a total of 941 Gb of raw data and produced 109 to 124 million sequence reads per group (Table 1). Over 97.6% of the generated sequence reads mapped to the annotated pigeon reference genome (Cliv_2.1), indicating that high-quality sequences were obtained. The average sequencing depth was 16.4× per group, within a range of 15.3- to 17.2-fold.

### 3.2. Identification and Annotation of Variants

A total of 15.06 M SNPs and 1.02 M InDels were detected in the mapped reads across all 53 samples, and the detailed information of each breed is shown in Table 2, with CR exhibiting the highest number of SNPs. Among the SNPs shared by the three groups, 8.56 M were SNPs, and the numbers of private SNPs in DZ, TRM, and CR were 1.18 M, 1.13 M, and 1.66 M, respectively (Figure 2).

### 3.3. Population Genetics

Neighbor-joining tree analysis (Figure 3a) and PCA (Figure 3b) supported three of the separate clusters, namely, DZ, TRM, and CR.

### 3.4. Selective Sweep Signals for the Taihu Dianzi Pigeon Special Piebalding

Using TRM and CR as the control groups, 66 and 69 selected regions, respectively, were identified by *F_ST_* and *π* ratio analyses in the DZ (Figure 4a,b), and 73 and 104 selected genes, respectively, were annotated in these regions. Furthermore, 57 genes were identified by overlap analysis (Figure 5, Appendix A), among which melanocortin-1 receptor (*MC1R*), enriched in the melanogenesis KEGG pathway, was associated with the special plumage color of the Taihu Dianzi pigeon. We identified two mutations in the *MC1R* gene in exon 1, including a synonymous mutation c. 279G>A and a missense mutation c. 520A>G (S174G), which were significantly correlated with the unique piebalding of the Taihu Dianzi pigeon (Figure 6).

### 3.5. Selective-Sweep Signals in Crested Pigeons

Using DZ1, DZ3, TRM, and CR as the control groups, 21, 28, 79, and 74 selected regions were identified by FST and π ratio analyses in DZ2 (Figure 7a–d), and 3, 12, 57, and 84 selected genes were annotated in these regions, respectively. However, no gene was identified by overlap analysis (Figure 8). Perhaps due to poor annotation of the pigeon genome, many genes were not annotated, preventing screening of selected genes.

### 3.6. Selective-Sweep Signals in Polydactyl Pigeons

Using DZ1, DZ2, TRM, and CR as the control groups, 32, 44, 72, and 68 selected regions were identified by *F_ST_* and *π* ratio analyses in DZ3 (Figure 9a–d), and 4, 6, 88, and 103 selected genes were annotated in these regions, respectively. Furthermore, three genes affecting the development of polydactyly, angiopoietin 4 (*ANGPT4*), histocompatibility minor 13 (*Hm13*), and solute carrier family 52 member 3 (*SLC52A3*), were identified by overlap analysis (Figure 10).

### 3.7. Genome-Wide Association Analysis

In order to further screen for genes that affect the crest and polydactyly traits, we conducted a GWAS. Six SNPs significantly correlated with the crest trait were identified through GWAS (Figure 11, Table 3). Among them, five significant loci were located at AKCR02000006.1. Unfortunately, only one gene, SET and MYND domain containing three (*SMYD*), near the SNP located at AKCR02000002.1 was annotated, and no genes near these SNPs located at AKCR02000006.1 were annotated due to a lack of sufficient annotation information on the pigeon genome. Therefore, we extracted the sequence covering the five SNPs at AKCR02000006.1 via alignment with the BLAST program (https://www.ncbi.nlm.nih.gov/ (accessed on 20 May 2023)) and inferred that the storkhead box two (*STOX2*) gene was contained in this region.

A total of 151 SNP loci significantly correlated with the polydactyly trait were identified through GWAS (Figure 12, Appendix A), and five genes, R-spondin 4 (*RSPO4*), tensin-3 (*TNS3*), chloride channel 7 (*CLCN7*), *SLC52A3*, and *ANGPT4*, were annotated to be significantly correlated with polydactyly. This result, combined with the results of the selective sweep, revealed two genes, *SLC52A3* and *ANGPT4*, that were enriched in riboflavin metabolism and the VEGF signaling pathway, respectively. Furthermore, one mutation in exon 1 of the *SLC52A3* gene (c. 222G>A) and three mutations in the 5’UTR (c.-169G>A), exon 5 (c. 786G>A), and exon 8 (c. 1172T>C) of the *ANGPT4* gene were associated with polydactyly. The detailed results are shown in Table 4. The mutations in these two genes may be related to the development of polydactyly in pigeons.

## 4. Discussion

The pigeon is one of the most well-known birds worldwide; it was probably domesticated at least 3000 years ago and possibly even earlier [23,24]. China is also a country with a long history of pigeon domestication, but there is a lack of identification and utilization of local pigeons, and the main commercial varieties rely on imports. To develop and utilize Chinese native pigeons, it is important to carry out molecular identification of local pigeons. The remarkable diversity of pigeons can be viewed as the outcome of massive selection activity. Pigeons exhibit dramatic variation in craniofacial structure; plumage color; placement and structure; foot features; vocalizations; flight ability; and many other traits [1]. Human-mediated selection can accelerate changes in animal morphological characteristics. Domestic pigeons undergo genomic variation under long-term natural and artificial selection pressures, which can directly affect the traits of their offspring [25]. The genomic data could reflect a wide variety of historical developments, including species introductions and artificial selection [26]. The exploration of population genetic structure and genetic diversity can be regarded as essential for genetic evaluation, reflecting cross-breeding and the utilization of genetic resources [27]. In this study, the neighbor-joining tree and PCA analysis showed that the three breeds were clustered by breed. Based on whole-genome sequencing data and combined with bioinformatics analysis, revealing the selection signals of important genetic traits remaining in the genome during animal domestication has become a mainstream approach [28,29,30]. Selection signals are the traces left on the genome of animals through long-term natural and artificial selection processes during domestication and typically manifest as linkage disequilibrium and reduced polymorphism at certain sites on both sides of the core variation [31]. The *F_ST_* and *π* ratio has been proven to be very an effective method for identifying selection and elimination areas, especially when mining functional areas closely related to special traits, and often produce a strong selection signal [32,33].

In this study, a comparative analysis between the Taihu Dianzi pigeon and two other pigeon breeds revealed that the *MC1R* gene region was strongly selected. MC1R is a G protein-coupled receptor expressed in cutaneous and hair follicle melanocytes and plays a central role in coat color determination in vertebrates [34]. Numerous mutations in this gene have been widely reported to be related to specific skin, coat, and plumage colors in animals [35,36,37]. Similarly, *MC1R* variations are also associated with the diversity of pigeon plumage colors [38]. Here, we identified two mutations, c. 279G>A and c. 520A>G in *MC1R* that are most likely responsible for the formation of the special piebalding black head and tail in pigeons. This result was consistent with the previous findings of our research [39].

The crest is a cluster of feathers protruding from the head that are widely present in various birds, but the morphology of the crest varies among different birds. Research on crested chickens has shown that the *HOX* genes play a role in the formation of the altered skull morphology related to the crest phenotype [40]. The gene controlling the chicken crest trait may be located in the *HOXC* cluster adjacent to the *HOXC8* gene [41]. The excess adipose tissue behind the cerebellum of the duck head affects the growth of the skull, and the high expression of the *HOXC8* and *EphA2* genes in intracranial adipose tissue may be related to the crest trait [42,43]. Mutations in the *TAS2R40* gene identified through GWAS may also affect the formation of duck crests [10]. There are many types of crests on pigeons. After whole-genome sequencing of a variety of pigeons with crests, Shapiro revealed that a SNP in the *EphB2* gene was significantly related to the crest trait [11]. This kind of crest is formed by the reverse growth of the head feathers during development, thus forming a crest at the back of the skull. However, the *EphB2* gene, which may not be the key gene involved in the formation of crests in the Taihu Dianzi pigeon, was not screened in this study. Although we did not identify genes related to the crest trait through selective sweeps, we identified six SNPs significantly associated with the crest trait through GWAS. However, whether *SMYD* and *STOX2* are the key genes controlling the crest trait of the Taihu Dianzi pigeon needs further experimental verification.

Polydactyly is a common limb abnormality in vertebrates. Extensive research has been conducted on polydactyly traits in humans [44], mice [45], and chickens [13], and various types of polydactylies have been classified. Pigeons are commonly four-toed, while the polydactyl Taihu Dianzi pigeons have extra toes on the outside of the inner toe of one or two feet, which is similar to the preaxial polydactyly [46]. Polydactyly in chickens occurs mainly on the short arm of chromosome 2, which harbors many candidate genes, such as *SHH*, *Lmbr1*, and *Gli3* [47,48]. At present, there are no reports of polydactyly in pigeons. In this study, two candidate genes, *SLC52A3* and *ANGPT4*, were identified through whole-genome resequencing combined with selective sweeps and GWAS. These genes may play important roles in the development of the polydactyly trait in the Taihu Dianzi pigeon.

The selection of many genes by selective sweep between the Taihu Dianzi pigeon and the other two pigeon breeds may be associated with the crest and polydactyly but was missed due to the high threshold set for subsequent screening. For example, the *PTCH1* gene, an essential part of hedgehog signaling, is associated with human polydactyly [49,50]. In addition, many of the selected genes may be related to other interesting traits of Taihu Dianzi pigeons, such as *CDH1* and *Gas8*, which are associated with facial features and need further exploration and analysis in the future [32]. Unfortunately, due to the limited degree of assembly and annotation of the pigeon genome, many identified SNPs cannot be precisely mapped, and related gene annotations cannot be performed. These specific results require further assembly and annotation of the pigeon genome.

## 5. Conclusions

Based on genome-wide resequencing data, combined with *F_ST_* and *π* ratio for system selection signal detection and GWAS for significant SNP association analysis, the genes that affecting the special phenotypes of the Taihu Dianzi pigeon were identified. The *MC1R* gene was related to the formation of the distinctive piebald of the Taihu Dianzi pigeon. *SMYD* and *STOX2* genes were related to the formation of the crest of the Taihu Dianzi pigeon. *SLC52A3* and *ANGPT4* genes might play an important role in the formation of the polydactyly of the Taihu Dianzi pigeon. The results of this study can provide a theoretical basis for the study of the pigeon piebalding, crest, and polydactyly traits.

## Figures and Tables

**Figure 1 animals-14-01047-f001:**
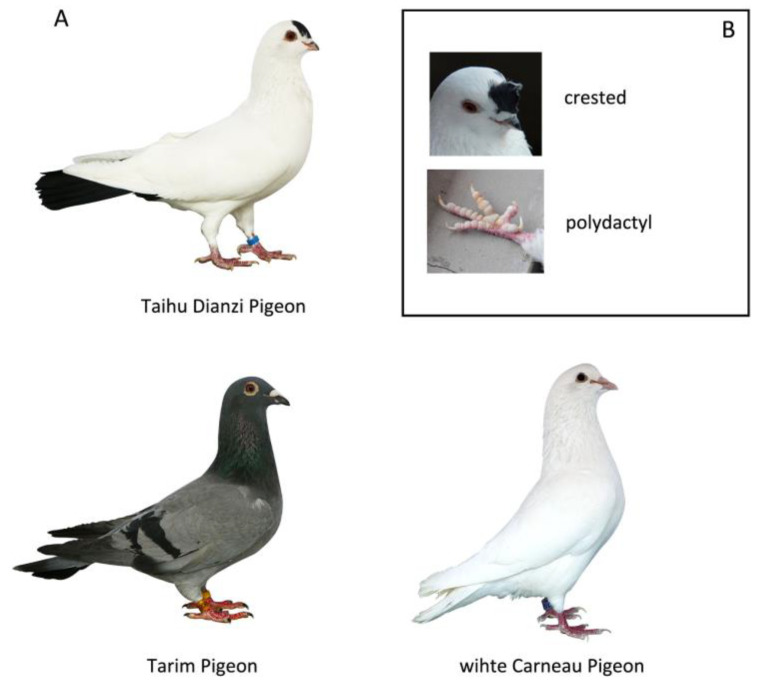
Representative images of the two native Chinese pigeon breeds and the introduced pigeon breed. (**A**) The Taihu Dianzi and Tarim pigeons are two native pigeon breeds in China. The white Carneau pigeon is an introduced pigeon breed from the United States. (**B**) Some of the Taihu Dianzi pigeons exhibit a crest or polydactyly.

**Figure 2 animals-14-01047-f002:**
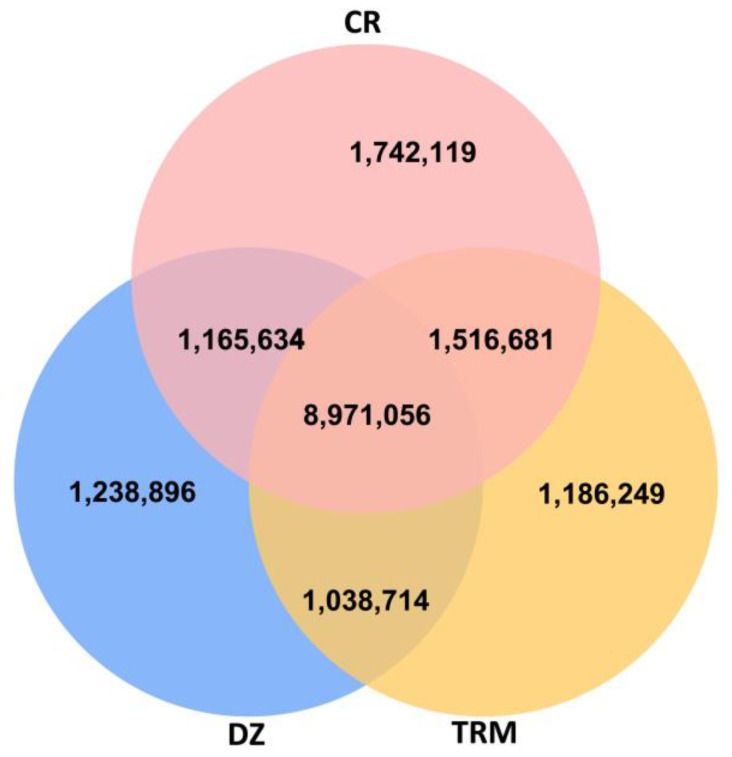
Venn diagram showing counts of shared and breed-specific variants in each breed. Taihu Dianzi pigeons (DZ), Tarim pigeons (TRM) and white Carneau pigeons (CR).

**Figure 3 animals-14-01047-f003:**
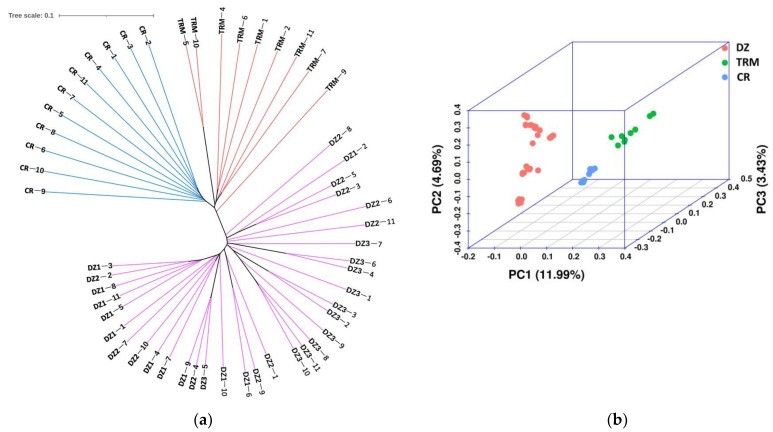
(**a**) Phylogenetic tree of the three pigeon breeds; (**b**) 3D PCA plot of the pigeon population.

**Figure 4 animals-14-01047-f004:**
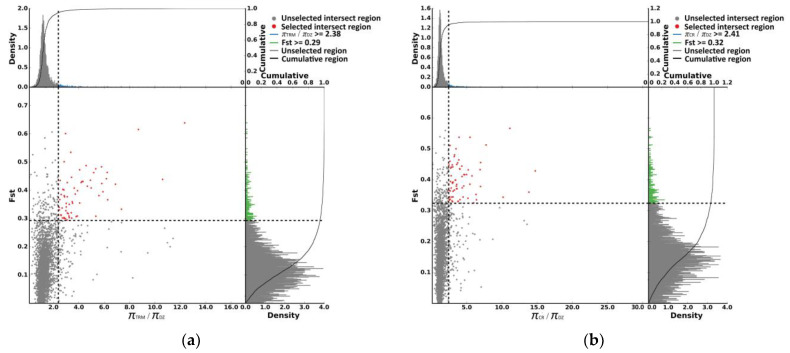
(**a**) The selective signals for parameter combinations of the *π* ratio (*π*TRM/*π*DZ) and *F_ST_* based on the top 5%; (**b**) The selective signals for parameter combinations of the π ratio (*π*CR/*π*DZ) and *F_ST_* based on the top 5%. The scatter plot represents the position of the coordinate plane determined by the values of two indicators.

**Figure 5 animals-14-01047-f005:**
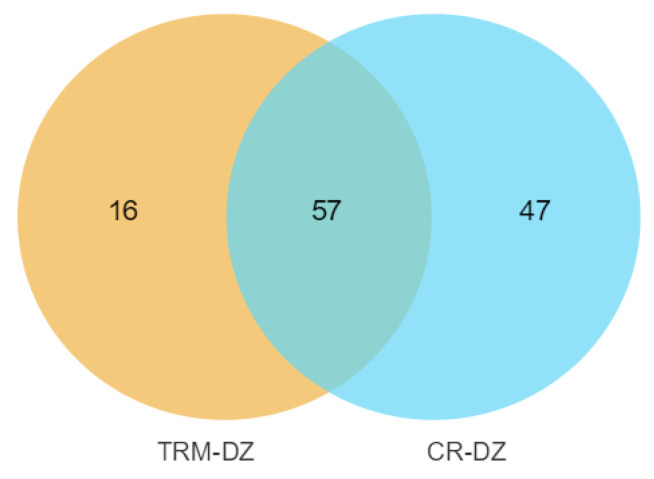
Venn diagram showing the number of overlapping candidate genes between DZ and the other two pigeons. TRM-DZ: The selected genes were identified by *F_ST_* and *π* ratio analyses in the DZ using TRM as the control groups. CR-DZ: The selected genes were identified by *F_ST_* and *π* ratio analyses in the DZ using CR as the control groups.

**Figure 6 animals-14-01047-f006:**
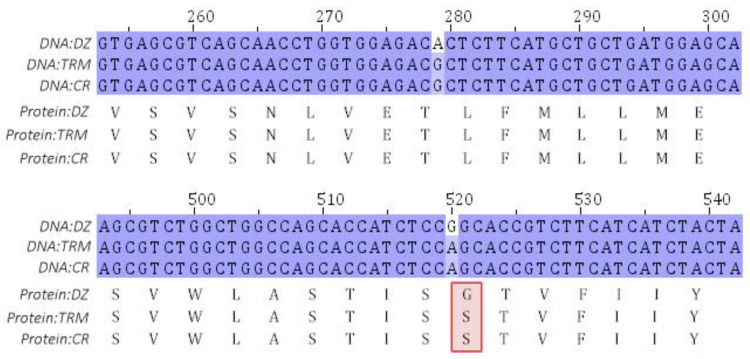
Alignment of nucleotide and amino acid sequences of *MC1R* among the three pigeon breeds. The same nucleotides are indicated in purple. Different amino acids are highlighted in the red box.

**Figure 7 animals-14-01047-f007:**
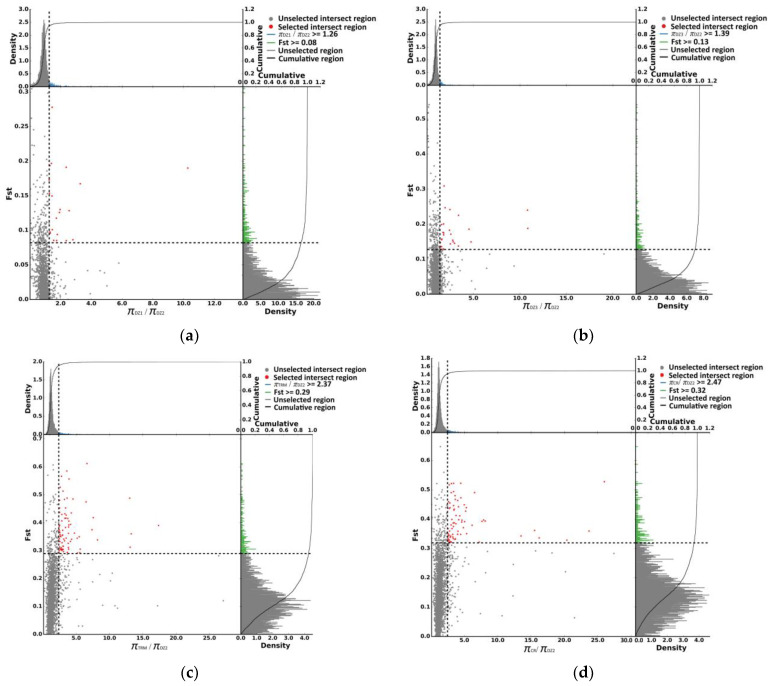
(**a**) The selective signals for parameter combinations of the *π* ratio (πDZ1/*π*DZ2) and *F_ST_* based on the top 5%. (**b**) The selective signals for parameter combinations of the *π* ratio (*π*DZ3/*π*DZ2) and *F_ST_* based on the top 5%. (**c**) The selective signals for parameter combinations of the π ratio (*π*TRM/*π*DZ2) and *F_ST_* based on the top 5%. (**d**) The selective signals for parameter combinations of the π ratio (*π*CR/*π*DZ2) and *F_ST_* based on the top 5%. The scatter plot represents the position of the coordinate plane determined by the values of two indicators.

**Figure 8 animals-14-01047-f008:**
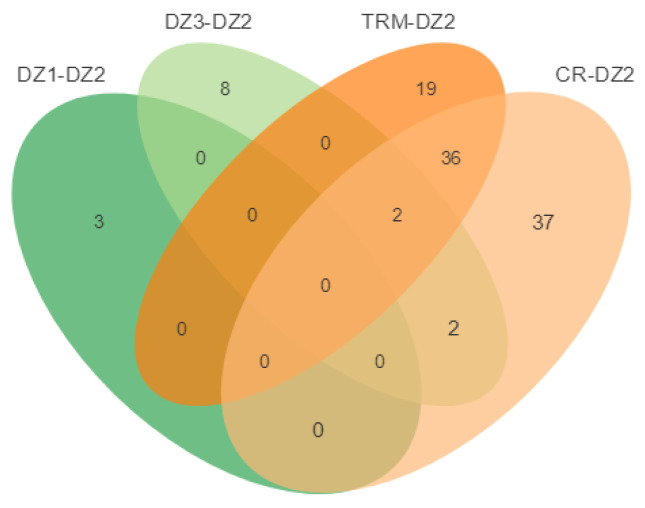
Venn diagram showing the number of overlapping candidate genes between DZ2 and the other groups. DZ1-DZ2: The selected genes were identified by *F_ST_* and *π* ratio analyses in the DZ2 using DZ1 as the control groups. DZ3-DZ2: The selected genes were identified by *F_ST_* and *π* ratio analyses in the DZ2 using DZ3 as the control groups. TRM-DZ2: The selected genes were identified by *F_ST_* and *π* ratio analyses in the DZ2 using TRM as the control groups. CR-DZ2: The selected genes were identified by *F_ST_* and *π* ratio analyses in the DZ2 using CR as the control groups.

**Figure 9 animals-14-01047-f009:**
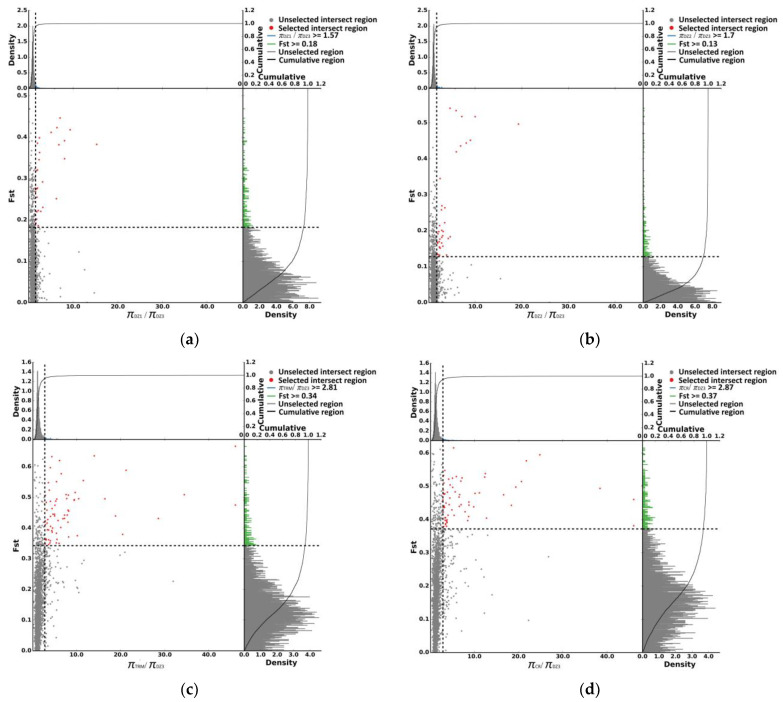
(**a**) The selective signals for parameter combinations of the *π* ratio (πDZ1/*π*DZ3) and *F_ST_* based on the top 5%. (**b**) The selective signals for parameter combinations of the *π* ratio (*π*DZ2/*π*DZ3) and *F_ST_* based on the top 5%. (**c**) The selective signals for parameter combinations of the π ratio (*π*TRM/*π*DZ3) and *F_ST_* based on the top 5%. (**d**) The selective signals for parameter combinations of the π ratio (*π*CR/*π*DZ3) and *F_ST_* based on the top 5%. The scatter plot represents the position of the coordinate plane determined by the values of two indicators.

**Figure 10 animals-14-01047-f010:**
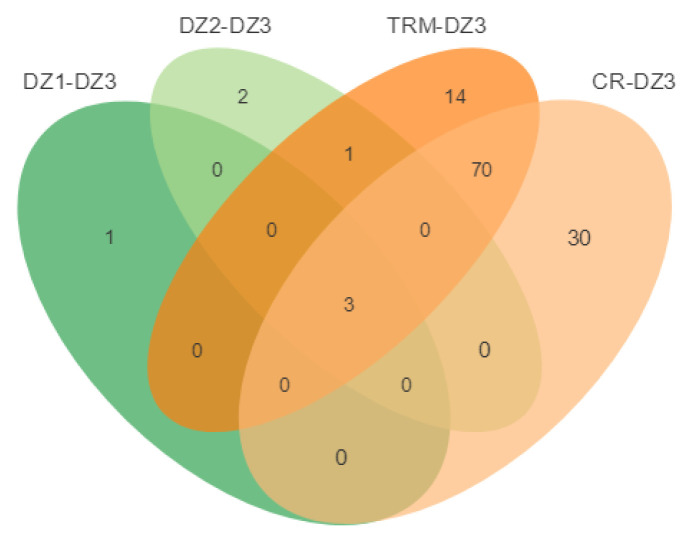
Venn diagram showing the number of overlapping candidate genes between DZ3 and the other groups. DZ1-DZ3: The selected genes were identified by *F_ST_* and *π* ratio analyses in the DZ3 using DZ1 as the control groups. DZ2-DZ3: The selected genes were identified by *F_ST_* and *π* ratio analyses in the DZ3 using DZ2 as the control groups. TRM-DZ3: The selected genes were identified by *F_ST_* and *π* ratio analyses in the DZ3 using TRM as the control groups. CR-DZ3: The selected genes were identified by *F_ST_* and *π* ratio analyses in the DZ3 using CR as the control groups.

**Figure 11 animals-14-01047-f011:**
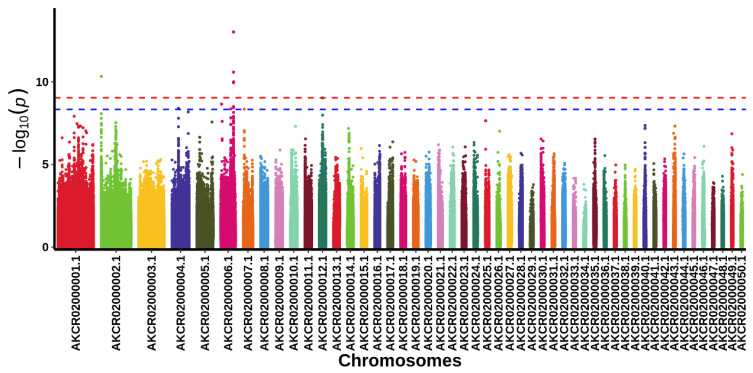
Manhattan plot of the genome-wide association analysis for the crest trait in pigeons. The *x*-axis shows the physical position of the SNPs by chromosome, and the *y*-axis shows −log10 (*p* values). The red dotted line indicates a highly significant genome-wide association (*p* < 0.01) according to the Bonferroni correction. The blue dotted line indicates a significant genome-wide association (*p* < 0.05) according to the Bonferroni correction.

**Figure 12 animals-14-01047-f012:**
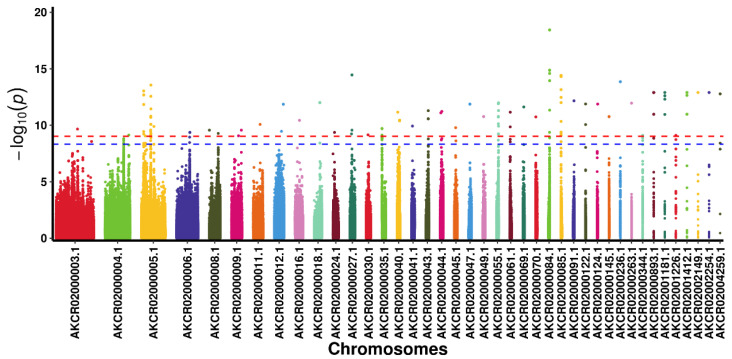
Manhattan plot of the genome-wide association analysis for the polydactyly trait in pigeons. The *x*-axis shows the physical position of the SNPs by chromosome, and the *y*-axis shows −log10 (*p* values). The red dotted line indicates a highly significant genome-wide association (*p* < 0.01) according to the Bonferroni correction. The blue dotted line indicates a significant genome-wide association (*p* < 0.05) according to the Bonferroni correction.

**Table 1 animals-14-01047-t001:** Summary of sequencing and short read alignment results.

Sample	N	Raw Data (G)	Clean Data (G)	Mapping Rate (%)	Sequence Depth (x)
DZ	33	18.26	17.71	97.60	17.16
TRM	9	17.74	17.18	97.77	16.64
CR	11	16.27	15.80	97.63	15.30

**Table 2 animals-14-01047-t002:** Functional annotation of the detected SNPs and InDels.

		DZ	TRM	CR
SNP	Total number of SNPs	11,578,452	11,866,346	12,516,118
	Intergenic	6,424,331	6,593,463	6,939,604
	Intronic	4,591,330	4,697,697	4,966,006
	Exonic	24,179	24,456	26,186
	Non-synonymous	42,855	42,599	46,243
	Stop gain	385	332	386
	Stop loss	46	49	58
	Synonymous	91,583	93,845	99,427
	Upstream	135,942	139,623	147,456
	Downstream	134,886	137,653	145,885
	UTR	84,396	86,490	92,215
	Splicing	491	434	469
	ncRNA	48,028	49,705	52,183
InDel	Total number of InDels	835,848	846,354	879,372
	Intergenic	456,527	462,451	478,820
	Intronic	344,699	349,138	364,232
	Exonic	348	347	348
	Non-synonymous	626	640	665
	Stop gain	12	12	13
	Stop loss	2	1	2
	Synonymous	545	556	577
	Upstream	10,641	10,676	11,073
	Downstream	12,134	12,173	12,694
	UTR	6528	6581	6971
	Splicing	171	178	187
	ncRNA	3615	3601	3790

**Table 3 animals-14-01047-t003:** The information of the SNPs significantly correlated with the crest trait.

SNP	Chr	Location	Mutation	*p*
SNP1	AKCR02000002.1	505029	T>G	4.7727 × 10^−11^
SNP2	AKCR02000006.1	33844338	G>A	1.1403 × 10^−10^
SNP3	AKCR02000006.1	33896741	C>T	1.0102 × 10^−10^
SNP4	AKCR02000006.1	33898464	G>A	9.6488 × 10^−14^
SNP5	AKCR02000006.1	33898620	C>T	9.6488 × 10^−14^
SNP6	AKCR02000006.1	33901426	C>A	2.6008 × 10^−14^

**Table 4 animals-14-01047-t004:** The association analysis between the SNPs in the candidate genes and polydactyly.

Gene	SNP	*p*	Genotype	Phenotype
SLC52A3	exon1:c.222G>A	1.2529 × 10^−13^	GG	non-polydactyly
			GA	non-polydactyly
			AA	polydactyly
ANGPT4	5’UTR:c.-169G>A	1.2529 × 10^−13^	GG	non-polydactyly
			GA	non-polydactyly
			AA	polydactyly
	exon5:c.786G>A	1.2529 × 10^−13^	GG	non-polydactyly
			GA	non-polydactyly
			AA	polydactyly
	exon8:c.1172T>C	1.2529 × 10^−13^	TT	non-polydactyly
			TC	non-polydactyly
			CC	polydactyly

## Data Availability

The data presented in this study are available in Appendix A.

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
