# Peer review of "Whole-Genome Sequencing for Identifying Candidate Genes Related to the Special Phenotypes of the Taihu Dianzi Pigeon"

_animals, 2024, doi:10.3390/ani14071047_

Round 1
Reviewer 1 Report
Comments and Suggestions for Authors
In Table 1, the sum of the SNPs and InDel counts does not match the total numbers presented. If the Authors apply any additional filters or exclusions, these must be explicitly detailed in the figure or table legends for clarity and transparency.
Similarly, the cumulative totals from the Venn diagram in Figure 2 do not correspond with the aggregate numbers detailed in Table 1. This discrepancy should be addressed, either within the figure legend or the main text, to ensure the accuracy and reliability of the data presented.
Regarding the legends of Venn diagrams in Figures 5, 8, and 10, the term "Overlapping candidate gene" is overly broad and lacks specificity. I recommended including information about the comparison or the related phenotypes.
Comments on the Quality of English LanguageMinor editing of English language required
Reviewer 2 Report
Comments and Suggestions for Authors
The authors should pay attention to:
Species scientific names - no scientific name is given to which species were used
Line 74: bred not cultivated
Figure 1: typing error
How many birds were crested and or polydactyl?
Population genetics results not discussed
Piebalding trait is not described in the introduction, only abstract, results and conclusion.
It is a pity the assembly of the genome wasn't good enough.
Reviewer 3 Report
Comments and Suggestions for Authors
The study of Zhang et al. aimed to genome-wide analysis and identification of candidate genes associated with piebald, crest, and polydactyly of Taihu Dianzi Pigeon. Pigeons are the most common birds in the world, but we still know little about their genome and the genes that determine the specific features of these birds. The work is interesting and well-written. Individual chapters have the correct structure. The work is innovative, but I have a few comments.
Introduction:
L38-L41: Please add citations regarding various characteristics of pigeons.
Figure 1.B. “polytactyl” is a correct word?
Materials and method:
Could you explain whether groups of 11, 9, and 33 individuals were representative for the study?
Please provide citations or producers for the all software you used.
Results:
(c.-169G>A) is this entry correct?
Reviewer 4 Report
Comments and Suggestions for Authors
The manuscript present a study on three pigeon breeds including two Chinese native breeds. The Taihu Dianzi pigeon, native to China, shows special characteristics such as piebald colour, crest formation and polydactyly.
Selective sweeps were analysed using the fixation index (FST) and nucleotide diversity (π) ratio and furthermore, GWAS.
MC1R was related to piebalding in Taihu Dianzi pigeons. Candidate genes for the crest (SMYD and STOX2) and polydactyly (SLC52A3 and 25 ANGPT4) were found.
The manuscript is well written and contains a lot of new data and informations.
Typos have to amended.
Figure 9 is hard to read.
Abbreviation used in figures should be explained in the legend.
The GWAS result for polydactyly is rather uncommon. Authors should check if something went wrong because so many signficiant loci on many different chromosomes look very strange. Did you not correct for the animal polygenic effect. Did you test DZ1 vs DZ3 and were other populations also included.
Comments on the Quality of English Language
No comments
Round 2
Reviewer 4 Report
Comments and Suggestions for Authors
The auhtors regarded all comments and amended their manuscript accordingly.
Some minor remarks
Line 15-16: Some candidate genes were identified by selective sweep analyses and GWAS.
Line 351: SLC52A3 and ANGPT4 genes might be played an important role in the formation of the 351 polydactyly of the Taihu Dianzi pigeon. >>>>> The SLC52A3 and ANGPT4 genes may have played an important role in the development of 351 polydactyly in the Taihu-Dianzi pigeon.
Spaces are missing after the brackets.
Comments on the Quality of English LanguageNo comments.